# Fish Consumption and Risk of Stroke in Chinese Adults: A Prospective Cohort Study in Shanghai, China

**DOI:** 10.3390/nu14204239

**Published:** 2022-10-12

**Authors:** Shuheng Cui, Kangqi Yi, Yiling Wu, Xuyan Su, Yu Xiang, Yuting Yu, Minhua Tang, Xin Tong, Maryam Zaid, Yonggen Jiang, Qi Zhao, Genming Zhao

**Affiliations:** 1Key Laboratory of Public Health Safety of Ministry of Education, Department of Epidemiology, School of Public Health, Fudan University, Shanghai 200032, China; 2Songjiang District Center for Disease Control and Prevention, Shanghai 201600, China

**Keywords:** fish consumption, stroke, prospective cohort study

## Abstract

Present studies on the association of fish consumption with risk of stroke have shown controversial results, and this association within the Chinese population remains unknown. We aimed to investigate the association between fish consumption and incidence of total stroke, ischemic stroke and hemorrhagic stroke among adults in China. We analyzed the data of 57,701 adults aged 20–74 years, with no history of stroke, in a prospective cohort study in Shanghai. Fish consumption was calculated from a food frequency questionnaire at baseline and divided into four categories (less than 300, 300–450, 450–600 and more than 600 g/week). Participant information was linked to health information systems in which stroke event information was collected up until 31 December 2021. The hazard ratios (HR) and 95% confidence intervals (CI) of the associations of fish consumption with risk of total stroke, ischemic stroke and hemorrhagic stroke were estimated using cox proportional hazards regression models. Dose–response relationships were estimated using restricted cubic spline analyses. During a median follow-up of 4.56 years, 807 newly developed stroke events were ascertained, including 664 ischemic stroke events and 113 hemorrhagic stroke events. Fish consumption of 300–450 g/week was associated with a reduced risk of total stroke (HR: 0.78, 95% CI: 0.64–0.94) and ischemic stroke (0.70 (0.57–0.88)) compared with fish consumption of less than 300 g/week, after adjustment for comprehensive covariates including sociodemographic characteristics, lifestyle, dietary patterns and disease histories. No significant association was found between fish consumption and hemorrhagic stroke. The findings of our study support the consumption level of fish recommended in the dietary guidelines.

## 1. Introduction

Fish is an important food source containing diverse nutrients such as selenium, vitamin B, vitamin D, and high-quality protein, as well as long-chain ω-3 polyunsaturated fatty acids (PUFAs), all of which have been shown to be beneficial to cardiovascular status [1,2,3,4,5,6]. In interventional studies, PUFAs and fish intake were suggested to reduce blood pressure and triglycerides levels [6,7,8]. Moreover, randomized clinical trials as well as observational studies have also shown that the moderate intake of PUFAs and fish reduced the risk of cardiovascular diseases (CVD) [9,10,11,12]. However, studies on the relationship of fish consumption and risk of stroke showed controversial results [12]. One meta-analysis of clinical trials showed that ω-3 supplementation was not associated with risk of stroke (summary relative risk (SRR): 1.05, 95% CI: 0.93–1.18) [6,13]. By contrast, another meta-analysis of prospective cohort studies indicated that each 100 g/d increment of intake of fish produced a lower risk of total stroke (SRR: 0.86, 95% CI: 0.75–0.99) [14]. One reason for this inconsistency could be that previous studies had different methodologies [12]. Importantly, the amounts and types of fish consumption differ across regions and populations. So far, most of the longitudinal studies on the association between fish consumption and incidence of stroke have been carried out in the United States, Japan and Europe. However, people consume various amounts and types of fish in China. One cohort in Guangzhou showed that the moderate consumption of fish was significantly associated with reduced risks of CVD mortality (HR and 95% CI: 0.85 (0.76–0.95) and 0.77 (0.64–0.93), respectively), but the association of fish consumption and incidence of stroke in the Chinese population remains unknown [15].

The excessive consumption of fish may result in excessive dietary fats. Therefore, a moderate consumption of fish is suggested in most of dietary guidelines. For example, the American Dietary Guidelines (2015–2020) recommend a consumption of about 8 ounces/week of seafood, and the Chinese Dietary Guidelines (2022) recommend a fish consumption of 300–500 g/week. With regard to the above-mentioned considerations, we aim to investigate the association of fish consumption with incidence of total stroke, ischemic stroke and hemorrhagic stroke in China. We hypothesized that moderate fish consumption was associated with lower risk of stroke in a Chinese population. In this study, we analyzed the association of fish consumption with incidence of stroke among adults in Shanghai in a prospective cohort. As far as we know, this is the largest cohort so far to be investigated for this association in China.

## 2. Materials and Methods

### 2.1. Study Design and Population

This cohort study consisted of the Shanghai Suburban Adult Cohort and Biobank (SSACB) study and a cohort study in Minhang district, Shanghai. Details of the SSACB have been described in our previous report [16]. In brief, four communities in Songjiang district and three in Jiading district were chosen firstly in terms of population size and economic status. One-third of the neighborhoods or villages in each community were then randomly selected as the study sites of the SSACB. Similar to the sampling methods of SSACB, three communities in Minhang district were selected, and one-third of the neighborhoods or villages in community were chosen to be the study sites of the Minhang cohort. Residents aged 20 to 74 years who have lived in Shanghai for at least 5 years were included according to their willingness to participance.

A total of 62,841 participants were recruited at baseline between 6 June 2016 and 27 October 2019. Participants attended face-to-face interviews where information on sociodemographic characteristics (including age, sex, socioeconomic status, marital status, retirement), medical histories, and aspects of lifestyle (including smoking, alcohol intake, tea drinking, physical activity, sleep and diet) were recorded by well-trained staff using standardized structured questionnaires. Anthropometric parameters (including height, weight, blood pressure and waist circumference) were measured with standardized methods by clinicians in community hospitals. Blood and urine samples were collected after 8 h of fasting and sent for biochemical testing. The reliability of the interview was tested by checking 5% of the recording files, which were randomly selected. Every participant was linked to their health record in the Cardiovascular and Cerebrovascular Disease Registration and Reporting System (CCDRR), the Electronic Medical Record System (EMR), and the Cause-of-Death Surveillance System (CDSS), with unique identification numbers. After the exclusion of 536 participants with incomplete data on fish consumption, 354 with incomplete or implausible data on critical variables including blood pressure, dietary energy intake and weight, and 4250 participants who already had histories of cancer, stroke, myocardial infarction or liver cirrhosis at baseline, a total of 57,701 participants were included in the present analysis (Figure 1). The study was approved by the Ethical Review Committee of the School of Public Health, Fudan University (IRB approval number 2016-04-0586). Written informed consent was obtained from all participants.

### 2.2. Assessment of Diet

A food frequency questionnaire (FFQ) including 29 categories of foods was used to assess the dietary intake at baseline, which recorded how often the participant had consumed each item of food on average in the last twelve months. There were eight frequency categories (never, less than 1 time/month, 1–3 times/month, 1–3 or 4–6 times/week, and 1, 2 or more than 3 times/day). An album of pictures of food portions was used to assess the amount of food intake. Dietary intake of fish is assessed via three categories in the FFQ, including freshwater fish (crucian carp, perch, catfish, carp, grass carp, etc.), marine fish (hairtail, pomfret, small yellow croaker, large yellow croaker, salmon, turbot, cod, etc.) and shrimp, crab and shellfish (river prawn, shrimp, river crab, sea crab, clam, etc.). The participants were divided into one of four categories according to fish consumption (less than 300 g/week, 300–450 g/week, 450–600 g/week, or more than 600 g/week).

### 2.3. Follow-Up and Ascertainment of Stroke

Follow-up was performed based on the linkage to health information systems including CCDRR, EMR and CDSS, which had detailed records on the names and dates of disease diagnoses, causes of death and dates of death. All diagnoses and causes were coded according to the International Classification of Diseases, Tenth Revision (ICD-10) [17]. A stroke event was defined as rapidly developing clinical signs of disorder of cerebral function lasting more than 24 h or leading to death (ICD-10 I60 to I69) [18]. An ischemic stroke event was defined as an episode of neurological dysfunction due to focal cerebral, spinal, or retinal infarction (ICD-10 I63) [18]. Intracerebral hemorrhage was defined as stroke with a focal collection of blood in the brain not due to trauma [18]. Subarachnoid hemorrhage was defined as non-traumatic stroke due to bleeding into the subarachnoid space of the brain [18]. A hemorrhagic stroke event was defined as a confirmed diagnosis of intracerebral hemorrhage or subarachnoid hemorrhage (ICD-10 I60 to I62).

### 2.4. Assessment of Covariates

Age, sex, educational degree, retirement, occupation, marital status, smoking status and alcohol intake were determined by self-report. Body mass index (BMI) was calculated by dividing the weight (kilograms) by height (meters) squared and divided into three groups (normal weight: BMI < 24.0 kg/m^2^; overweight: 24.0 ≤ BMI ≤ 28.0 kg/m^2^; obesity: BMI ≥ 28 kg/m^2^) [19]. Hypertension was defined as systolic blood pressure (SPB) ≥ 140 mmHg or diastolic blood pressure (DBP) ≥ 90 mmHg, or previous diagnosis with hypertension [20,21]. Coronary heart disease (CHD), chronic obstructive pulmonary disease (COPD), chronic bronchitis and asthma were defined according to disease histories. Diabetes was defined as fasting blood glucose (FPG) ≥ 7.0 mmol/L, glycosylated hemoglobin, type A1c (HbA1c) ≥ 6.5% or previous diagnosis with diabetes [22]. Chronic kidney disease (CKD) was defined as a persistent abnormality in kidney function or kidney impairment, including the estimated glomerular filtration rate (eGFR) < 60 mL/min per 1.73 m^2^, hematuria or proteinuria [23]. Dyslipidemia was defined as total cholesterol ≥ 6.20 mmol/L, high-density lipoprotein cholesterol (HDL-C) < 1.00 mmol/L, low-density lipoprotein cholesterol (LDL-C) ≥ 4.10 mmol/L or triglyceride ≥ 2.30 mmol/L [24]. Hyperuricemia (HUA) was defined as serum uric acid ≥ 420 μmol/L in males and ≥ 360 μmol/L in females [25]. Hyperhomocysteinemia (HHcy) was defined as serum homocysteine (Hcy) > 15 μmol/L [26]. Physical activities (PA) were assessed as the metabolic equivalent tasks (METs) multiplied by the total number of minutes per week (METs-min/week) based on the International Physical Activity Questionnaire (IPAQ) and divided into tertiles (low: <3089; moderate: 3090–5040; and high: > 5040) [27]. Sleep quality was calculated as the score of the Pittsburgh Sleep Quality Index (PSQI) and divided into tertiles [28]. Dietary energy intake, and the consumption of fruit, vegetables, peanuts, whole grains, processed and unprocessed meats, bean products, salt and oil, were all calculated from the FFQ and divided into groups according to the Chinese Dietary Guidelines (2022) when adjusted in the models (Appendix A).

### 2.5. Statistical Analyses

Estimation of sample size was performed with PASS 15 software (NCSS, LLC, Kaysville, UT). We used tests for two proportions to estimate the sample size. The estimated incidence rates of total stroke, ischemic stroke and hemorrhagic stroke were 350, 250 and 50 per 100,000 person years, respectively [18]. The test used a two-sided Z-Test with pooled variance. The significance level of the test was 0.05, and the power to detect an HR of 0.70 was 90.00%. In the result, the sample sizes we needed for total stroke, ischemic stroke and hemorrhagic stroke were 119,362, 166,997 and 833,866 person years, assuming 5.00% of loss to follow-up.

The baseline characteristics of participants across four categories of fish consumption are shown by the median values, with the interquartile range (IQR) for continuous variables not normally distributed, and as percentages for categorical variables. The Kruskal–Wallis rank test was used for continuous variables and the Mantel–Haenszel χ^2^ test was used for categorical variables. The Cochran–Armitage trend test was used to test the incidence density trend of stroke events across four categories of fish consumption. The HR and 95% CI of stroke and subtypes was estimated using cox proportional hazards regression models. We adjusted for age at baseline, sex, educational levels, marital status and retirement status in model 1. In model 2, smoking status, alcohol drinking status, physical activity levels, sleep qualities, obesity status, dietary energy intakes and consumption of fruit, vegetables, peanuts, wholegrains, processed and unprocessed meets, bean products, salt and oil were further adjusted. In Model 3, we additionally adjusted for histories of chronic diseases, including hypertension, CHD, CKD, dyslipidemia, diabetes, HUA, HHcy, COPD, chronic bronchitis and asthma. The categories and weighted percentages of covariates in the models are presented in Appendix A. We treated the median values in each category as continuous variables to test the linear trend.

Stratified analyses were subsequently conducted to evaluate whether the association differed by age (<60 years, ≥60 years), sex, smoking status (current smokers or not), alcohol drinking status (current alcohol users or not), BMI groups (<24, ≥24 kg/m^2^), history of hypertension (yes or no), history of diabetes (yes or no) and history of dyslipidemia (yes or no). We tested the interaction between those variables and fish consumption by including a cross-product term in the model. Dose–response association was investigated by restricted cubic spline analyses. In sensitivity analyses, we excluded deaths within the first two years of follow-up. We also conducted an analysis after excluding the category of shrimp, crab and shellfish.

Our study conformed to STROBE guidelines. SAS, version 9.4 (SAS Institute, Inc., Cary, NC, USA) was used in all statistical analyses. All the tests were two-tailed, and *p* < 0.05 was considered statistically significant.

## 3. Results

### 3.1. Baseline Characteristics and the Incidence of Stroke

The baseline characteristics of 57,701 participants across four categories of fish consumption are presented in Table 1. In total, 39.71% of the participants were male, and the median age of participants was 59 (51–65) years. The median fish consumption of participants was approximately 300 g/week (180–550 g/week). Participants of higher fish consumption were younger, more likely to be men, current smokers, current drinkers and current tea drinkers, having higher educational levels and dietary energy intakes, with higher consumption levels of vegetables, peanuts, bean products and unprocessed meats, and they were less likely to have histories of hypertension, chronic bronchitis and CKD. Over a total of 248,263.39 person years’ follow-up (with the median follow-up duration of 4.56 years), we ascertained 807 newly developed stroke events, including 664 cases of ischemic stroke, 113 cases of hemorrhagic stroke and 30 cases of unspecified stroke. The incidence densities and 95% CIs of total stroke events, ischemic stroke and hemorrhagic stroke were 325.10 (303.20–348.10), 267.50 (247.70–288.40) and 45.52 (37.69–54.51) per 100,000 person years, respectively. Participants who consumed 300–450 g/week of fish had the lowest incidence density of total stroke and ischemic stroke, while participants who consumed 450–600 g/week of fish had the lowest incidence density of hemorrhagic stroke, as is shown in Figure 2.

### 3.2. Association of Fish Consumption with Stroke

As shown in Table 2, fish consumption of 300–450 g/week had beneficial effects in total stroke and ischemic stroke compared to the consumption of less than 300 g/week, based on model 3. The HRs and 95% CIs were 0.78 (0.64–0.94) and 0.70 (0.57–0.88), respectively. Fish consumption of 450–600 g/week reduced the risk of total stroke and ischemic stroke compared to fish consumption of less than 300 g/week in model 1 (HRs and 95% CIs were 0.77 (0.62–0.96) and 0.78 (0.61–0.99), respectively). However, the association was no longer significant in further adjusted models. There was no association found between fish consumption of more than 600 g/week and risk of any type of stroke, nor any association of fish consumption with risk of hemorrhagic stroke. Linear trends were not found (all *p* values for trends were >0.05 in model 3).

In stratified analyses (Figure 3, Figure 4 and Figure 5), no effect modification was observed resulting from age (<60 years, ≥60 years), sex, smoking status, alcohol drinking status, BMI groups, history of hypertension, diabetes or dyslipidemia (all *p* values for interaction > 0.05). Fish consumption of 300–450 g/week reduced the risk of total stroke in those aged ≥ 60 years, men, current smokers, non-current alcohol users, BMI ≥ 24, and with histories of hypertension, diabetes and dyslipidemia. Fish consumption of 450–600 g/week lowered the risk of total stroke in those aged ≥ 60, who were current smokers and had histories of diabetes (Figure 3). Fish consumption of 300–450 g/week reduced the risk of ischemic stroke in participants aged ≥ 60 years, men, current alcohol users, BMI ≥ 24, and those with a history of hypertension or diabetes, and the association was significant regardless of whether they were current smokers or had histories of dyslipidemia. Fish consumption of 450–600 g/week reduced the risk of ischemic stroke only in those aged ≥ 60 years and current smokers (Figure 4). No association was found between consumption of more than 600 g/week of fish and risk of stroke in any of the subgroups, nor of fish consumption with hemorrhagic stroke (Figure 5).

### 3.3. Dose–Response Analysis of Fish Consumption with Stroke

There was a nonlinear association of fish consumption with total stroke and ischemic stroke, as found by restricted cubic spline analyses (all *p* for nonlinear trend < 0.05; Figure 6A,B), while no such nonlinear associations were observed for hemorrhagic stroke (*p* for nonlinear trend > 0.05; Figure 6C).

The results remained consistent even when excluding deaths within the first two years of follow-up (Appendix A). We also excluded the categories of shrimp, crab and shellfish. Because the total amount of fish consumption became smaller after excluding the category of shrimp, crab and shellfish, and the number of participants who consumed more than 600 g/week of fish became too small, we adjusted the category of fish consumption into four new groups (less than 150 g/week, 150–300 g/week, 300–450 g/week, or more than 450 g/week). The results remained consistent; fish consumption of 300–450 g/week reduced the risk of total stroke and ischemic stroke compared with consumption of less than 150 g/week (Appendix A).

## 4. Discussion

This prospective cohort study among adults in Shanghai showed that fish consumption of 300–450 g/week reduced the risk of total stroke and ischemic stroke compared to fish consumption of less than 300 g/week. As far as we know, this is the largest study at present to examine the association between fish consumption and incidence of stroke in China. Our results support the consumption level of fish recommended in the Chinese Dietary Guidelines (2022).

Our study shows similar results to those of some previous studies. A meta-analysis of 15 prospective studies indicated that an increment of three servings/week in fish consumption reduced the risk of total stroke and ischemic stroke but had no effect on hemorrhagic stroke (SRR and 95% CI: 0.94 (0.89–0.99, 0.90 (0.84–0.97), and 0.90 (0.76–1.06), respectively) [9]. Another pooled analysis of four cohort studies involving 191,558 individuals suggested that fish consumption of 175–350 g/week reduced the risk of total stroke compared with less than 50 g/month (HR and 95% CI: 0.81 (0.72–0.92)), but the associations with subtypes of stroke were not reported [6]. There were also some studies that showed inconsistent results. A prospective study among 34,033 Dutch adults showed no association between fish consumption (compared to non-consumption) and incidence of stroke [29]. Another study in a Spanish cohort of the European Prospective Investigation into Cancer and Nutrition (EPIC-Spain) observed no significant association of fish consumption with the risk of stroke both in men and women [30]. Different populations, lifestyles, eating habits of fish, cooking methods, and levels of polychlorinated biphenyls and other environmental contaminants in fish could explain the different results between studies [31]. In China, people consume a considerable amount of freshwater fish and crabs, while Western populations consume a larger proportion of marine fish. As for marine fish, Chinese residents consume more hairtail, pomfret and yellow croaker than salmon, turbot, cod or herring, which are more commonly consumed in Western countries. Fish is mostly steamed in China but is often fried in the United States. Frying would increase the energy density of fish and lift the amounts of advanced glycation end products, which could increase the risk of stroke [32]. In Northern Europe, marine fish such as herring and salmon are usually pickled with salt and sugar, while high salt consumption could lift the blood pressure and elevate the risk of stroke [12,33]. Thus, cooking habits may account for some of the inconsistent results across different studies on the association between fish consumption and stroke.

Fish is the main dietary source of PUFAs, which may decrease the risk of stroke through the following mechanisms. First of all, PUFAs could reduce blood pressure [31]. Second, a high consumption of fish oil could reduce triglyceride levels in serious hypertriglyceridemia patients [7,8]. Clinical trials have illustrated that two servings of fatty fish per week (approximately 112 g per serving) reduced triglyceride levels by 11.4%, but also lifted LDL-C levels slightly compared to the control group [34,35]. The increase in LDL-C levels may explain why a high fish consumption of more than 600 g/week no longer has beneficial effects on the risk of stroke, but it does not raise the risk of stroke due to the beneficial effects of lowering triglyceride and neutralizing the associated risks [36]. Moreover, PUFAs have beneficial effects, preventing platelet aggregation [37] and oxidative stress [38], and could lower blood viscosity and increase arterial compliance [39].

In our study, the association of moderate fish intake with reduced risk of total stroke and ischemic stroke was more pronounced in men, which is compatible with a nested case–control study conducted in Northern Sweden [40]. Some risk factors of stroke might have different distributions in males and females. However, the results were unchanged after adjustments for lifestyle factors including smoking, alcohol drinking, physical activities, sleep quality, and diet habits concerning a series of food types.

Our study has some advantages. It has a very large population with robust data, a prospective design, and complete linkages to health information systems, which ensure the accurate ascertainment of stroke events and subtypes without loss to follow-up, adjustment for comprehensive covariates, and diverse sensitivity analyses to examine the stability of the results. However, it also has some limitations. First, the duration of follow-up is relatively short, and the number of hemorrhagic stroke events was not enough to draw a significant conclusion. As is shown in the estimation of sample size, a follow-up of at least 833,866 person years is required for enough hemorrhagic stroke events. Therefore, a long-term follow-up is required for further examination of the association between fish consumption and hemorrhagic stroke. Second, although the FFQ we used was designed for Chinese residents whose fish consumption habits differ from other populations in terms of the type of fish and amounts consumed, it has not been validated. Therefore, the validation of the FFQ is required. Third, the diet was only recorded once at baseline. Therefore, we did not take changes in diet over time into account. However, considering that the follow-up period was relatively short (median: 4.56 years), it is likely that participants had stable dietary habits. Moreover, we did not determine the associations of different types of fish with stroke outcomes. Finally, residual confounding was inevitable in our study, as is the case with all observational studies, even though we controlled for comprehensive covariates including sociodemographic characteristics, lifestyle, dietary pattern and disease history.

## 5. Conclusions

In summary, fish consumption of 300–450 g/week reduced the risk of total stroke and ischemic stroke compared with fish consumption of less than 300 g/week among adults in Shanghai. These findings support the consumption level of fish recommended in dietary guidelines. We call for further research distinguishing different types of fish and a longer follow-up period.

## Figures and Tables

**Figure 1 nutrients-14-04239-f001:**
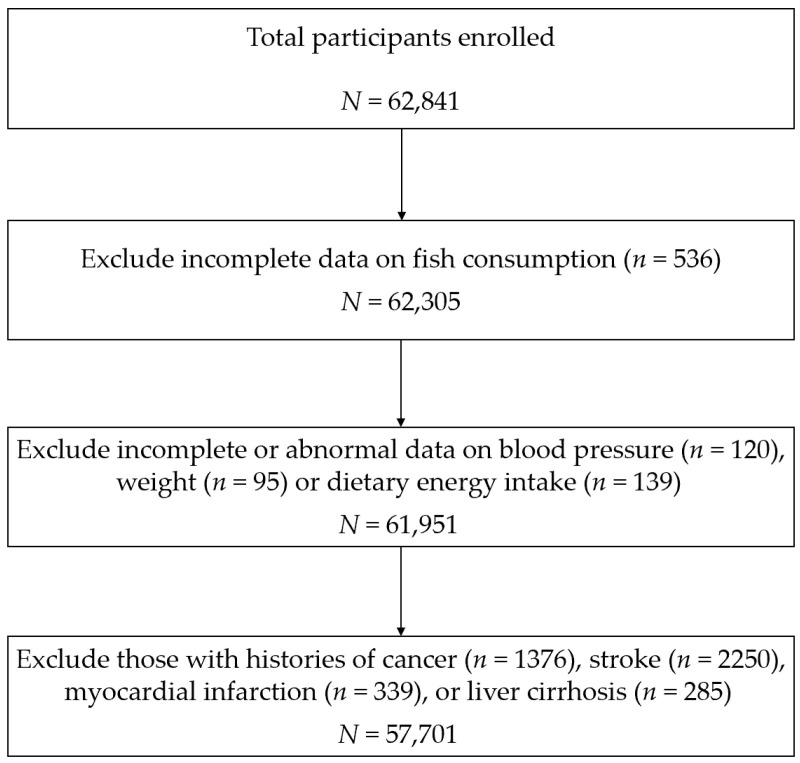
Flow of the participant screening.

**Figure 2 nutrients-14-04239-f002:**
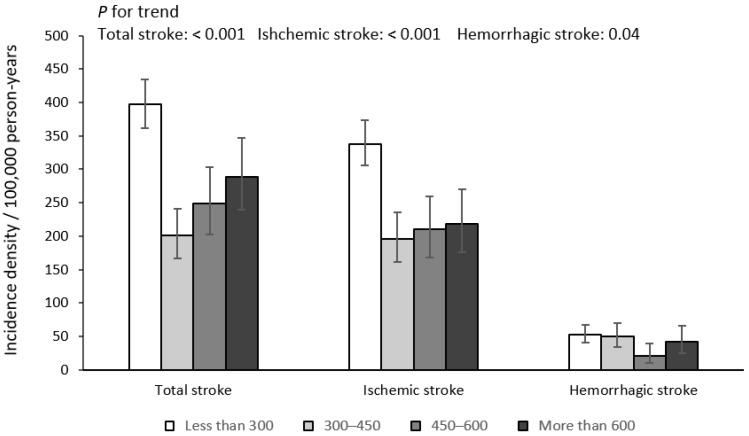
Incidence densities of total stroke, ischemic stroke and hemorrhagic stroke across four categories of fish consumption (g/week).

**Figure 3 nutrients-14-04239-f003:**
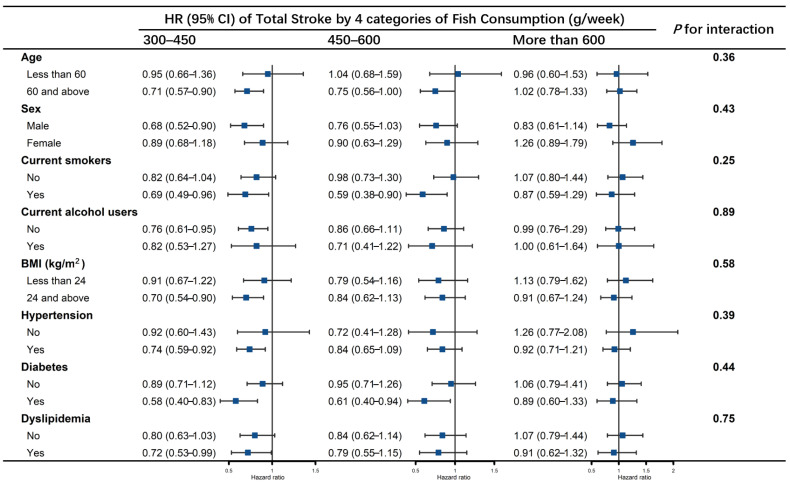
Association of fish consumption (g/week) between the risk of total stroke by subgroups of age, sex, smoking status, alcohol drinking status, BMI groups, history of hypertension, diabetes, and dyslipidemia. The model was adjusted for age, sex, educational levels, marital status, retirement status, smoking status, alcohol drinking status, physical activity levels, sleep qualities, obesity status, dietary energy intakes, consumption of fruit, vegetables, peanuts, wholegrains, processed and unprocessed meats, bean products, salt and oil, histories of chronic diseases, including hypertension, CHD, diabetes, CKD, dyslipidemia, HUA, HHcy, COPD, chronic bronchitis and asthma (the same as covariates in Model 3). Each subgroup analysis was adjusted for all the covariates listed above except itself.

**Figure 4 nutrients-14-04239-f004:**
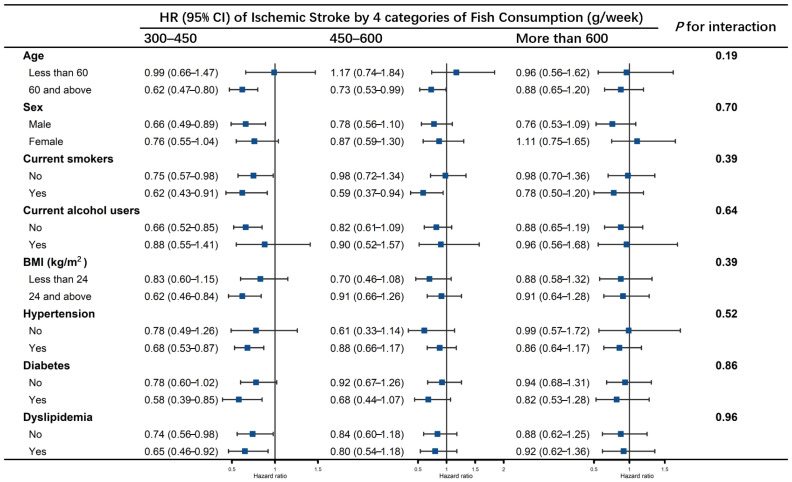
Association of fish consumption (g/week) between the risk of ischemic stroke by subgroups of age, sex, smoking status, alcohol drinking status, BMI groups, history of hypertension, diabetes, and dyslipidemia. The model was adjusted for age, sex, educational levels, marital status, retirement status, smoking status, alcohol drinking status, physical activity levels, sleep qualities, obesity status, dietary energy intakes, consumption of fruit, vegetables, peanuts, wholegrains, processed and unprocessed meats, bean products, salt and oil, histories of chronic diseases, including hypertension, CHD, diabetes, CKD, dyslipidemia, HUA, HHcy, COPD, chronic bronchitis and asthma (the same as covariates in Model 3). Each subgroup analysis was adjusted for all the covariates listed above except itself.

**Figure 5 nutrients-14-04239-f005:**
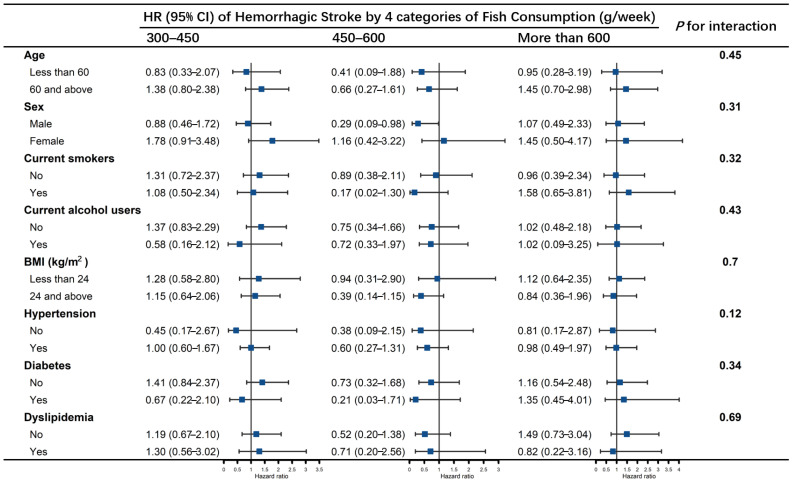
Association of fish consumption (g/week) between the risk of hemorrhagic stroke by subgroups of age, sex, smoking status, alcohol drinking status, BMI groups, history of hypertension, diabetes, and dyslipidemia. The model was adjusted for age, sex, educational levels, marital status, retirement status, smoking status, alcohol drinking status, physical activity levels, sleep qualities, obesity status, dietary energy intakes, consumption of fruit, vegetables, peanuts, wholegrains, processed and unprocessed meats, bean products, salt and oil, histories of chronic diseases, including hypertension, CHD, diabetes, CKD, dyslipidemia, HUA, HHcy, COPD, chronic bronchitis and asthma (the same as covariates in Model 3). Each subgroup analysis was adjusted for all the covariates listed above except itself.

**Figure 6 nutrients-14-04239-f006:**
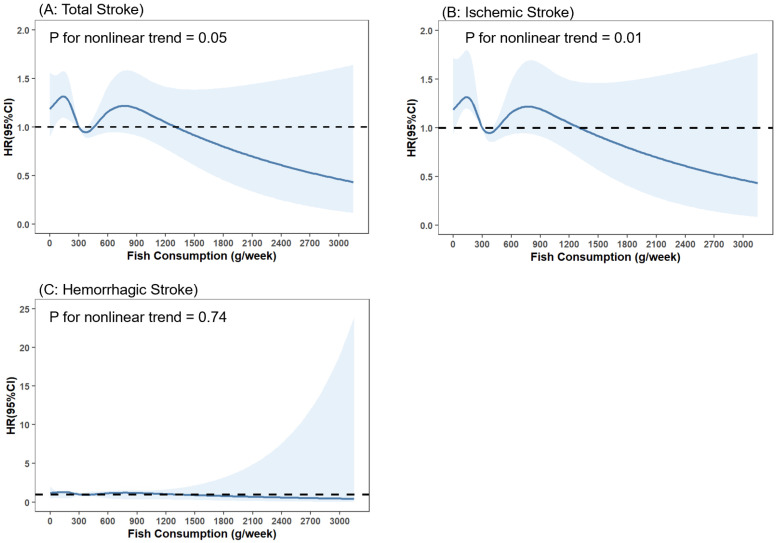
Association of fish consumption (g/week) with total stroke (**A**), ischemic stroke (**B**) and hemorrhagic stroke (**C**) in restricted cubic spline models. The model was adjusted for age, sex, educational levels, marital status, retirement status, smoking status, alcohol drinking status, physical activity levels, sleep qualities, obesity status, dietary energy intakes, consumption of fruit, vegetables, peanuts, wholegrains, processed and unprocessed meats, bean products, salt and oil, histories of chronic diseases, including hypertension, CHD, diabetes, CKD, dyslipidemia, HUA, HHcy, COPD, chronic bronchitis and asthma (the same as covariates in Model 3).

**Table 1 nutrients-14-04239-t001:** Baseline characteristics of participants according to four categories of fish consumption.

Characteristics	Total (*n* = 57,701)	Fish Consumption (g/week)				*p*-Value
		Less than 300 (*n* = 25,977)	300–450 (*n* = 13,222)	450–600 (*n* = 9223)	More than 600 (*n* = 9279)	
Age (year)	59 (51–65)	60 (52–66)	58 (50–64)	57 (49–64)	57 (49–64)	<0.001
Men (%)	22,914 (39.71)	9780 (37.65)	5093 (38.52)	3895 (42.23)	4146 (44.68)	<0.001
Newly developed stroke (%)						
Total stroke	807 (1.40)	457 (1.76)	144 (1.09)	95 (1.03)	111 (1.20)	<0.001
Ischemic stroke	664 (1.15)	389 (1.50)	111 (0.84)	80 (0.87)	84 (0.91)	<0.001
Hemorrhagic stroke	113 (0.20)	61 (0.23)	28 (0.21)	8 (0.09)	16 (0.17)	0.04
Educational levels (%)						<0.001
Primary school or below	20,712 (35.90)	12,071 (46.47)	4232 (32.01)	2234 (24.22)	2175 (23.44)	
Junior high school	22,531 (39.05)	8924 (34.35)	5613 (42.45)	4053 (43.94)	3941 (42.47)	
Senior high school	8927 (15.47)	3038 (11.69)	2126 (16.08)	1821 (19.74)	1942 (20.93)	
High school or above	5531 (9.59)	1944 (7.48)	1251 (9.46)	1115 (12.09)	1221 (13.16)	
Marriage status (%)						<0.001
Married	53,455 (92.64)	23,715 (91.25)	12,375 (93.59)	8692 (94.24)	8683 (93.58)	
Unmarried	867 (1.50)	451 (1.74)	170 (1.29)	110 (1.19)	136 (1.47)	
Divorced and other	3379 (5.86)	1821 (7.01)	677 (5.12)	421 (4.56)	460 (4.96)	
Retired (%)	35,355 (61.27)	16,715 (64.35)	8006 (60.55)	5306 (57.53)	5328 (57.42)	<0.001
Smoking status (%)						<0.001
Never	44,642 (77.37)	20,438 (78.68)	10,331 (78.13)	7021 (76.12)	6852 (73.84)	
Former	2110 (3.66)	954 (3.67)	442 (3.34)	307 (3.33)	407 (4.39)	
Current	10,949 (18.98)	4585 (17.65)	2449 (18.52)	1895 (20.55)	2020 (21.77)	
Current alcohol users (%)	7216 (12.51)	3025 (11.64)	1550 (11.72)	1210 (13.12)	1431 (15.42)	<0.001
Current tea drinkers (%)	16,724 (28.98)	6559 (25.25)	3844 (29.07)	3044 (33.00)	3277 (35.32)	<0.001
PA level (%)						0.25
Low	19,233 (33.33)	8776 (33.78)	4180 (31.61)	3133 (33.97)	3144 (33.88)	
Moderate	19,211 (33.29)	8750 (33.68)	4436 (33.55)	3037 (32.93)	2988 (32.20)	
High	19,257 (33.37)	8451 (32.53)	4606 (34.84)	3053 (33.10)	3147 (33.92)	
PSQI score	3 (2–5)	3 (2–5)	3 (2–5)	3 (2–5)	3 (2–5)	<0.001
BMI (kg/m^2^)	23.94 (21.72–26.21)	23.98 (21.76–26.29)	23.92 (21.72–26.14)	23.91 (21.67–26.11)	23.86 (21.64–26.20)	<0.001
History of chronic diseases (%)						
Hypertension	36,468 (63.20)	16,764 (64.53)	8343 (63.10)	5667 (61.44)	5694 (61.36)	<0.001
CHD	2454 (4.25)	1187 (4.57)	520 (3.93)	389 (4.22)	358 (3.86)	0.003
Diabetes	8663 (15.01)	4163 (16.03)	1883 (14.24)	1288 (13.97)	1329 (14.32)	<0.001
CKD	8251 (14.30)	3989 (15.36)	1843 (13.94)	1235 (13.39)	1184 (12.76)	<0.001
Dyslipidemia	17,981 (31.16)	7967 (30.67)	4044 (30.59)	2986 (32.38)	2984 (32.16)	0.001
HUA	9692 (16.80)	4552 (17.52)	2069 (15.65)	1506 (16.33)	1565 (16.87)	0.03
HHcy	13,288 (23.03)	6070 (23.37)	2970 (22.46)	2054 (22.27)	2194 (23.64)	0.70
COPD	295 (0.51)	145 (0.56)	59 (0.45)	41 (0.44)	50 (0.54)	0.37
Chronic bronchitis	3797 (6.58)	1905 (7.33)	812 (6.14)	544 (5.90)	536 (5.78)	<0.001
Asthma	1203 (2.08)	607 (2.34)	271 (2.05)	150 (1.63)	175 (1.89)	<0.001
Energy intake (kcal/day)	1457.37 (1156.38–1831.30)	1283.41 (1028.47–1609.73)	1454.11 (1190.47–1769.58)	1610.22 (1330.03–1956.11)	1850.86 (1504.12–2332.94)	<0.001
Fruit (g/day)	80.00 (28.57–150.00)	50.00 (25.71–100.00)	100.00 (28.57–150.00)	100.00 (42.86–150.00)	100.00 (50.00–200.00)	<0.001
Vegetables (g/day)	242.86 (135.15–406.35)	206.58 (104.11–334.29)	242.86 (150.00–378.57)	300.00 (200.00–428.57)	350.00 (228.57–542.86)	<0.001
Peanuts (g/week)	23.03 (3.43–100.03)	23.03 (1.75–93.03)	39.97 (5.74–100.03)	46.06 (9.24–140.00)	46.06 (5.74–175.00)	<0.001
Whole grains (g/day)	6.58 (0.82–28.57)	5.14 (0–14.29)	6.58 (1.64–28.57)	14.29 (2.47–28.57)	14.29 (1.97–30.00)	<0.001
Unprocessed meats (g/day)	48.31 (28.58–85.15)	35.27 (20.87–59.87)	51.08 (35.15–79.65)	67.57 (42.87–106.58)	78.58 (45.01–128.99)	<0.001
Processed meats (eat %)	30,282 (52.48)	11,827 (45.53)	6409 (48.47)	4772 (51.74)	4411 (47.54)	<0.001
Bean products (g/day)	18.40 (6.58–41.72)	14.29 (3.29–28.57)	27.44 (14.29–41.72)	28.57 (14.29–57.15)	28.57 (14.29–71.43)	<0.001
Salt (g/day)	4.00 (2.67–6.25)	4.17 (2.67–6.67)	3.33 (2.67–5.56)	3.33 (2.67–5.56)	4.00 (2.67–6.40)	<0.001
Oil (g/day)	28.57 (20.00–41.67)	33.33 (20.00–41.67)	27.78 (20.00–41.67)	27.78 (20.00–41.67)	27.78 (18.52–41.67)	<0.001

Median (interquartile range) was displayed for continuous variables and percentages for categorical variables. Kruskal–Wallis rank test was used for continuous variables and Mantel-Haenszel χ^2^ test was used for categorical variables. PA, physical activities; METs, metabolic equivalents; PSQI, Pittsburgh Sleep Quality Index; CHD, coronary heart disease; CKD, chronic kidney disease; HUA, hyperuricemia; HHcy, hyperhomocysteinemia; COPD, chronic obstructive pulmonary disease.

**Table 2 nutrients-14-04239-t002:** Hazard ratios (95% CIs) of total stroke, ischemic stroke and hemorrhagic stroke by four categories of fish consumption.

	Fish Consumption (g/week)				*p* for Trend
	Less than 300	300–450	450–600	More than 600	
Total stroke					
Model 1	1.00	0.74 (0.61–0.89)	0.77 (0.62–0.96)	0.88 (0.72–1.09)	0.05
Model 2	1.00	0.77 (0.64–0.94)	0.82 (0.65–1.04)	1.00 (0.80–1.27)	0.55
Model 3	1.00	0.78 (0.64–0.94)	0.82 (0.65–1.04)	1.00 (0.79–1.26)	0.52
Ischemic stroke					
Model 1	1.00	0.68 (0.55–0.84)	0.78 (0.61–0.99)	0.80 (0.66–1.02)	0.01
Model 2	1.00	0.70 (0.56–0.87)	0.83 (0.64–1.07)	0.91 (0.70–1.18)	0.20
Model 3	1.00	0.70 (0.57–0.88)	0.82 (0.64–1.07)	0.90 (0.69–1.17)	0.18
Hemorrhagic stroke					
Model 1	1.00	1.08 (0.69–1.69)	0.49 (0.23–1.03)	0.96 (0.55–1.68)	0.47
Model 2	1.00	1.22 (0.77–1.96)	0.58 (0.27–1.25)	1.24 (0.67–2.32)	0.81
Model 3	1.00	1.21 (0.76–1.94)	0.58 (0.27–1.25)	1.24 (0.66–2.30)	0.82

We treated the median value in each category as continuous variables to test the linear trend. Model 1: adjusted for age, sex, educational levels, marital status and retirement status. Model 2: adjusted further for smoking status, alcohol drinking status, physical activity levels, sleep qualities, obesity status, dietary energy intakes and consumption of fruit, vegetables, peanuts, wholegrains, processed and unprocessed meats, bean products, salt and oil. Model 3: adjusted further for histories of chronic diseases, including hypertension, CHD, diabetes, CKD, dyslipidemia, HUA, HHcy, COPD, chronic bronchitis and asthma.

## Data Availability

The dataset used and analyzed during the current study is available from the corresponding author upon reasonable request.

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
