# Peer review of "Fish Consumption and Risk of Stroke in Chinese Adults: A Prospective Cohort Study in Shanghai, China"

_nutrients, 2022, doi:10.3390/nu14204239_

Round 1

Reviewer 1 Report

I read the current study with great interest. It has a very large population with robust data and it is well-written and well-designed.

I have some comments:

- I noticed that around 22.5% of population were current smokers. It would be important to add data about ex-smokers too.

- According to my above comment, among the comorbidities, data about respiratory diseases (e.g. asthma, COPD, etc.) should be added as well as data about neoplastic diseases.

- Did authors estimate sample size? Please specify it in data analysis section.

- I found some minor English error throughout the manuscript. Please have a deep language revision.

Reviewer 2 Report

The manuscript presented by Cui et al. entitled “Fish Consumption and Risk of Stroke in Chinese Adults: A Prospective Cohort Study in Shanghai, China “ investigated the association of fish consumption and incidence of stroke in adults in Shanghai.

The main finding of this work is that moderate fish intake was associated with reduced risk of total stroke and ischemic stroke. These results supported the consumption level of fish recommended in the Chinese Dietary Guidelines (2022).

I feel that the manuscript is interesting and well written, however, it needs minor improvement to be considered for publication.

1. The aim of the work should be better grasped in the abstract as well as in the introduction.

2. The paper requires English language editing.
